# Ecological Connectivity for Reptiles in Agroecosystems: A Case Study with Olive Groves in Liguria (Northwestern Italy)

**DOI:** 10.3390/ani15070909

**Published:** 2025-03-21

**Authors:** Andrea Costa, Fabrizio Oneto, Giacomo Rosa, Giacomo Actis Dato, Dario Ottonello

**Affiliations:** 1Department of Earth, Environmental and Life Sciences (DISTAV), University of Genova, Corso Europa 26, 16132 Genova, Italy; giacomo.rosa@edu.unige.it; 2Centro Studi Bionaturalistici (CeSBiN), Via San Vincenzo 2, 16121 Genova, Italy; 3Agenzia Regionale per la Protezione Dell’ambiente Ligure (ARPAL), Via Bombrini 8, 16149 Genova, Italy; dario.ottonello@arpal.liguria.it

**Keywords:** agroecosystems, connectivity, olive grove, reptiles, traditional land use practices

## Abstract

Agriculture is essential for food security but has significantly impacted biodiversity, particularly through habitat loss and fragmentation. In the Mediterranean region, traditional farming systems like olive groves have coexisted with nature for centuries, creating landscapes that support diverse wildlife. This study investigates whether olive groves help maintain ecological connectivity for reptiles in Liguria, Italy, a region with a long history of olive cultivation. Using a large dataset of reptile records and advanced modeling techniques, we assessed how olive groves influence reptile movement across the landscape. Our results show that olive groves do not act as barriers to reptile movement and provide connectivity levels nearly identical to those of natural habitats. Furthermore, olive groves often serve as corridors linking protected areas, facilitating reptile movement between these regions. These findings highlight the importance of preserving traditional olive groves as part of biodiversity conservation strategies, especially in human-modified landscapes. By promoting sustainable farming practices, we can balance agricultural production with the need to protect wildlife, ensuring healthier ecosystems for future generations.

## 1. Introduction

Agriculture is a pivotal practice for providing food security and economic stability to mankind, its importance even increasing recently, given the ongoing climate crisis and growing global population. However, as the largest form of land use on Earth, covering approximately 38% of the global land surface [1], agriculture and its associated expansion and intensification have become significant drivers of biodiversity loss and habitat fragmentation [2]. The conversion of natural ecosystems into agricultural landscapes often leads to the fragmentation of natural habitats and to the consequent degradation of ecological connectivity, which is critical for the conservation of entire biological communities [3,4,5]. Ecological connectivity impacts the survival of many species with limited dispersal capabilities, particularly those relying on narrow ecological requirements [6].

Despite this, not all agricultural practices are equally detrimental to biodiversity. In some regions, traditional and extensive farming systems have coexisted with nature for centuries, or even millennia, creating stable human-modified landscapes that can support viable biological communities [7]. The Mediterranean region, in particular, is a perfect example of this long-lasting interplay between human activities and natural ecosystems. For millennia, traditional agricultural practices—such as low-intensity pasturelands, terraced farming, water management systems, and Mediterranean olive groves—have shaped the landscape, creating a mosaic of semi-natural and semi-artificial habitats that support high biodiversity levels [7]. These landscapes are highly heterogeneous, offering a diverse array of microhabitats, microclimatic refugia, and shelters for various species [8,9,10]. Many organisms have adapted to these environments, which, even when entirely artificial—such as dry-stone walls or agricultural water tanks—can offer shelter [9,11] or sustain complex trophic webs [12]. Traditional olive groves, for instance, represent a long-lasting example of an artificial environment that is often managed with minimal chemical inputs, maintains a diverse understory vegetation, and can support a wide range of flora and fauna [13].

In addition to providing microclimatic refugia, mating opportunities, and trophic sites, semi-natural habitats in the Mediterranean cultural landscape can also contribute to biodiversity, maintaining ecological connectivity among remnant suitable patches within an unhospitable landscape [3,14]. Ecological connectivity refers to the degree to which landscapes facilitate or impede the movement of organisms and the flow of ecological processes [15]. It is a fundamental aspect of biodiversity conservation, as it enables species to access resources, find mates, and colonize new areas, thereby maintaining genetic diversity and population viability [4,16,17,18]. However, these landscapes also present unique opportunities for conservation, as traditional land use practices often create a network of semi-natural habitats that can function as ecological corridors, a feature of primary importance for ensuring ecological connectivity [17,18]. In human-modified landscapes, such as those found in the Mediterranean, maintaining ecological connectivity is particularly challenging due to the fragmentation of natural habitats caused by the shift from extensive to intensive agriculture and urban development, as well as land abandonment [19]. For example, hedgerows, riparian buffers, water management systems, and traditional farming systems like olive groves can enhance connectivity by linking patches of natural vegetation and facilitating species movement [20]. In the Mediterranean, where the landscape is a dynamic and complex mosaic of natural and human-modified elements, understanding and enhancing ecological connectivity is essential for preserving biodiversity in the face of the ongoing land use changes [21,22].

Reptiles, being ectothermic and often habitat-specialized, are highly sensitive to changes in land use and habitat structure [23,24]. Reptiles are among the most threatened groups of vertebrates globally, with habitat loss and fragmentation being major drivers of their current decline [24]. Agricultural intensification, which often involves the removal of natural vegetation, the use of pesticides added to the indirect loss of prey, and the simplification of landscape structure, has been shown to negatively impact biodiversity as a whole [25] and reptiles in particular [26]. However, low-intensity farming systems, which mimic natural habitats and maintain structural complexity, can provide suitable conditions for reptiles, indicating that semi-natural patches of traditional cultural landscape can support diverse reptile communities, particularly when they are embedded in a heterogeneous landscape with patches of natural vegetation (e.g., [11]). Olive trees, their understory, and olive groves in general are known to be used by reptiles for shelter [27] or trophic activity [28] and are usually capable of sustaining rich reptile communities, at least when compared to landscapes subjected to more intensive agricultural practices [29,30,31]. In this context, despite olive groves being recognized as suitable habitats for reptiles and possibly playing a crucial role in their conservation within fragmented landscapes, and also considering the known importance of single, large tree stands in providing tree-related microhabitats important for biodiversity [32,33], the contribution of olive groves to maintaining ecological connectivity at the landscape scale remains widely understudied.

The aim of this study was to evaluate the role of olive groves in landscape connectivity for the entire reptile community in Liguria, a region in NW Italy with a long-standing tradition of olive production and a high density of olive groves. In particular, our study aimed to address three interrelated questions regarding the role of olive groves in ecological connectivity:

Q1: Do olive groves act as barriers for reptile connectivity across the landscape?

Q2: When embedded within natural habitats, do olive groves provide the same connectivity as the surrounding natural patches?

Q3: What role do olive groves play in facilitating connectivity for reptiles between protected natural areas? In other words, do olive groves serve as a part of the ecological corridors connecting protected areas?

The findings of this study will have significant implications for reptile biodiversity conservation in agricultural landscapes. By evaluating the role of traditional olive groves in maintaining ecological connectivity, we aimed to provide insights that can guide land management practices, balancing agricultural production with biodiversity conservation. This is particularly relevant in the context of global efforts to achieve sustainable development goals, where understanding agriculture’s impact on biodiversity is crucial for designing landscapes that promote both human well-being and ecological integrity.

## 2. Materials and Methods

### 2.1. Study Rationale and Framework

To evaluate the contribution of olive groves to ecological connectivity for reptiles at the landscape scale, we developed the following analytical framework (Figure 1). We leveraged a large regional dataset, comprising 5211 georeferenced reptile records spanning from 2000 to 2024, which we used to build taxon-specific resistance maps, informed by habitat suitability models. These maps were then employed to build an omnidirectional connectivity map for the whole reptile community at the landscape scale by means of electric circuit theory modeling.

Finally, we analyzed the contribution of olive groves to reptiles’ landscape connectivity to answer the three questions presented previously.

### 2.2. Study Area and Reptiles’ Occurrence Data

The study area consists of the whole Liguria’s administrative area (Figure 2), covering a surface of 5418 km^2^. Liguria is a narrow coastal region in northwestern Italy, stretching along the Ligurian Sea and bordered by the Alps and Apennines. Its topography is characterized by steep slopes, valleys, and a rugged coastline, creating a diverse range of microhabitats. The region is mainly inhabited along the coastline, where urban and industrial development is prevalent, while the inner mountainous areas remain less populated. Liguria is characterized by a Mediterranean climate along its coastline, with mild, wet winters and warm, dry summers. Inland areas, due to their mountainous terrain, experience cooler temperatures and higher precipitation, influenced by orographic effects [34]. Liguria’s vegetation is dominated by Mediterranean maquis, mixed forests, and traditional agricultural landscapes, of which olive groves and terraced vineyards are the most representative [35]. 

The dataset, consisting of georeferenced reptile occurrences, used for the analyses was extracted from the regional biodiversity observatory (Li.Bi.Oss–Osservatorio Ligure della Biodiversità), an official open access database of the Ligurian region managed by the Agenzia Regionale per la Protezione dell’Ambiente Ligure (ARPAL) aimed at collecting, validating, and organizing occurrence data to track and monitor regional biodiversity. For our analyses, we excluded from the database pond turtles (*Emys orbicularis*) and allochthonous turtles (e.g., *Trachemys scripta*), since their occurrence is limited to water bodies and was not of interest in the present study. At the same time, we considered genetically close species (e.g., *Chalcides striatus* and *Chalcides chalcides*) as a single entity at the genus taxonomic level. The resulting dataset consisted of 5211 reptile records for 14 taxa, spanning from 2000 to 2024 (Figure 3; Table 1).

### 2.3. Building Landscape-Scale Connectivity Maps for Reptiles

In order to evaluate the landscape-scale contribution of olive groves to reptile ecological connectivity, particularly if the final aim is to inform systematic conservation planning frameworks [36], it is essential to assess the effectiveness of a management strategy for a broad range of species. Therefore, in this study, we decided to evaluate the contribution of olive groves to ecological connectivity using a “species-agnostic” approach [4], that is, evaluating the contribution of olive groves for the entire reptile community, instead of reporting species-specific results. Connectivity models, especially those built by means of electric circuit theory [5], are commonly employed to assess functional connectivity—i.e., how organisms move and respond to landscape structures—and to identify key areas or barriers for dispersal [4]. Electric circuit theory is widely applied to estimate the electrical flow between nodes on a resistance surface, where this flow represents dispersal and diffusion routes across different landscape resistances [5]. This approach simulates animal movement through random walks—a stochastic process generating a path composed of sequential random steps—along all potential movement pathways, with the intensity of the current flow between nodes corresponding to the frequency with which an individual traverses that route [5]. As a result, the current flow serves as an indicator of movement probability across the landscape, effectively measuring landscape connectivity [4,5]. Circuit theory analysis generates a map of electric current density, which represents the likelihood of a given pathway being used by considering all possible connections between nodes on a resistance surface [5]. Resistance maps play a crucial role in connectivity modeling, as they depict the level of friction or resistance that landscape features impose on animal movement. These maps can be constructed using expert-based assessments, where friction values are subjectively assigned to different landscape configurations or environmental features [37,38], or can be derived from analytical approaches informed by empirical data, such as habitat suitability models (HSMs) [3,36,37]. In this study, we chose to transform an HSM into a resistance surface by applying the formula (1 − suitability × 100). This approach has been successfully employed in the last years to study different biological systems: endangered amphibians subject to landscape and climate change [39], martens [40], leopards [41], Mediterranean amphibian communities [3], and amphibian pathogens [42]. Specifically, we built resistance maps starting from an HSM for each taxon included in the dataset, using Maxent (V 3.4.4), a machine learning algorithm that creates a model of habitat suitability based on gridded predictors at observed occurrence locations [43]. This approach is particularly performative when working with presence-only data [44], and therefore we consider it especially appropriate for reptile data, when absence data are not available or could be highly biased due to imperfect detection. For building the HSMs, we selected six environmental variables, with a spatial resolution corresponding to a mesh size of 100 m, capable of explaining reptile occurrence, which are reported in detail in Table 2.

HSMs were developed by sampling environmental conditions from 10,000 randomly selected background points. To prevent niche truncation, the models were extended beyond the study area, incorporating a buffer that encompassed the neighboring administrative regions [45]. Model performance was determined using the area under the receiver operating characteristic curve (AUC), considering that models with an AUC of ~0.75 have a good predictive ability [46]. Random sampling of 75% of the data was used to train the model, while the remaining 25% was used to test model performance. We repeated this step 10 times and averaged habitat suitability model predictions across replicated runs [46].

Circuit theory connectivity models have been usually built by using species occurrence locations or habitat patches as circuit nodes. However, this approach may be restrictive if the interest is to produce maps of landscape permeability in the whole study area [3,41,42]. To overcome this issue, we relied on omnidirectional methods, using a moving window algorithm, obtaining a landscape-scale connectivity map, unbiased by nodes placement [47]. For this purpose, we converted our HSMs to resistance surfaces (1 − suitability × 100) for modeling landscape connectivity. We employed the suitability values of every cell to identify the current sources, meaning that only the cells with suitability above a specific threshold value were used as sources [42], employing the maximum specificity plus sensitivity as a threshold value, a measure highly used for converting suitability values to presence/absence [48]. This approach is more conservative than injecting current from occurrence sites, which can be biased by unequal sampling efforts and imperfect detection. We built circuit theory connectivity models using the Omniscape connectivity algorithm implemented within the Julia programming environment [49]. Omniscape applies the circuit theory algorithm across a landscape, using a circular moving window, and evaluates the current flow between every possible pair of pixels within the moving window [49]. Subsequently, the resulting maps are summed to a cumulative map of current flow across the landscape. Omniscape provides three different outputs: (i) cumulative current flow (CCF), which is the total current flowing through the landscape; (ii) flow potential (FP), which represents a null model of current flow, assuming that movement is unconstrained by resistance (i.e., using a constant resistance map); (iii) normalized current flow (NCF; calculated as CCF/FP), which is the most representative metric of ecological connectivity and represents functional connectivity at the landscape scale. The NCF defines areas where current is impeded (i.e., movement is contrasted by the landscape; NCF < 1), diffuse (i.e., no resistance to movement; NCF ≈ 1), intensified (NFC > 1), or channeled (i.e., pinch points, where movement is constrained between barriers and is greater than expected by the null model; NFC >> 1) [47]. In order to provide community-level connectivity maps, our last step consisted in averaging taxon-specific NCF maps, obtaining a landscape-scale connectivity map for reptiles [50]. In the final step, recognizing that the study area encompassed distributional range boundaries for certain species (e.g., *Malpolon monspessulanus*, *Timon lepidus*, *Hierophis viridiflavus*), the connectivity layers for these species were clipped, so to align with their known distribution ranges. The areas outside these ranges were assigned a value of zero before proceeding with the averaging process.

### 2.4. Q1: Do Olive Groves Act as Barriers for Reptile Connectivity Across the Landscape?

In order to evaluate the effect of olive groves on the ecological connectivity of reptiles at the landscape scale, we overlaid a vector layer representing the land use/land cover categories for the study area (Land Use Map 2019, provided by Regione Liguria—https://geoportal.regione.liguria.it, accessed on 1 November 2024) onto the ecological connectivity map and extracted the values of normalized current flow, assessing the NCF values for the olive groves and considering that NCF values << 1 identify barriers to movement.

### 2.5. Q2: When Embedded Within Natural Habitats, Do Olive Groves Provide the Same Connectivity as the Surrounding Natural Patches?

To assess whether olive groves provide the same level of connectivity as the surrounding natural patches within which they are embedded, we followed a structured approach. First, we extracted the neighboring patches of each olive grove from the land cover/land use map. We then filtered for natural patches, retaining only those categories that represented at least 2% of the total area adjacent to olive groves within the study area. Finally, we calculated the NCF values for both olive groves and natural land cover categories and compared them using a Kruskal–Wallis test.

### 2.6. Q3: Do Olive Groves Serve as a Part of the Ecological Corridors Connecting Protected Areas?

To assess the role of olive groves in facilitating reptile connectivity between protected natural areas, we divided the study area into two regions: western and eastern Liguria, where most olive groves are located. We selected all protected areas within these regions (n = 32 for western Liguria and n = 29 for eastern Liguria; see Appendix A Appendix A) and calculated the least-cost paths connecting all pairs of protected areas within each region [51]. The inverse of the NCF was used as the cost surface (calculated as maxNCF − NCFi, where NCFi represents the NCF value of each cell). The least-cost paths were generated using the Least Cost Path plugin in QGIS V3.4. To optimize the computational efficiency, we reduced the number of potential links from each protected area by retaining only the five nearest target areas. Finally, we assessed the role of olive groves within ecological networks by evaluating the proportion of least-cost paths intersecting at least one olive grove.

## 3. Results

### 3.1. Landscape Scale Connectivity Maps for Reptiles

The maximum entropy suitability models, for all taxa, were characterized by a good predictive ability (Appendix A Appendix A), with AUC scores ranging between 0.764 (*Chalcides* sp.) and 0.934 (*Euleptes europaea*). Elevation above sea level and tree cover density were the most important variables for all taxa, with suitability that was generally negatively affected by elevation and positively associated with intermediate or low levels of tree cover density (Appendix A Appendix A). The landscape-scale ecological connectivity maps for the entire reptile community, obtained by omnidirectional electric circuit theory modeling, showed that the current flow was mainly diffuse, although some areas were characterized by an impeded current flow, and a few scattered areas presented an intensified current flow (Figure 4).

### 3.2. Q1: Do Olive Groves Act as Barriers for Reptile Connectivity Across the Landscape?

From the regional land use/land cover map, we extracted 4212 polygons, representing the amount of different olive groves present in the study area. The values of NCF for the olive groves indicated a current flow ranging from highly impeded to intensified (min = 0.12; max = 2.33). The average NCF value for the olive groves was 0.96 (SD = 0.12), corresponding to a diffuse current flow for this agroecosystem.

### 3.3. Q2: When Embedded Within Natural Habitats, Do Olive Groves Provide the Same Connectivity as the Surrounding Natural Patches?

The 4212 polygons representing olive groves neighbored 6185 polygons representing natural habitat patches. The most represented natural habitat patches were mixed thermophilic woodlands and transitional woodlands/shrub (Table 3). The average values of NCF for these habitat categories were all close to 1, indicating that reptile movement across this habitat categories was not impeded. Despite this, from the Kruskal–Wallis test, a significant difference emerged for the NCF values across all habitat categories (Kruskal–Wallis test; chi^2^ = 155.3; *p* < 0.001). Specifically, from a Mann–Whitney post hoc test after Bonferroni correction, the NCF values for the olive groves showed significant differences with respect to those recorded for transitional woodlands/shrubs (*p* < 0.01) and those characterized by sclerophyllous vegetation (*p* < 0.001) and chestnut woodlands (*p* < 0.001; Figure 5).

### 3.4. Q3: Do Olive Groves Serve as a Part of the Ecological Corridors Connecting Protected Areas?

Considering the two sub-regions of Liguria (western and eastern), this analysis identified 32 protected areas in the western sub-region and 29 protected areas in the eastern sub-region, which were connected by 145 and 160 least-cost paths, respectively, with a maximum of five links per protected area. Among these paths, 99 (61.9%) in the western sub-region and 124 (85.5%) in the eastern sub-region included at least one olive grove.

## 4. Discussion

The findings of our study provide the first insights into the role of olive groves in maintaining ecological connectivity for reptiles in Mediterranean landscapes. By addressing the three research questions posed in this article, we demonstrated that olive groves, a traditional agricultural system, play a nuanced but significant role in facilitating reptile movement across the landscape. These results may have important implications in the context of biodiversity conservation and sustainable land management.

### 4.1. Q1: Do Olive Groves Act as Barriers for Reptile Connectivity Across the Landscape?

Our analysis revealed that olive groves generally do not act as barriers to reptile movement across the landscape. The average normalized current flow (NCF) value for olive groves indicated a diffuse current flow that neither significantly impedes nor intensifies reptile movement [47]. This suggests that olive groves, despite being human-modified habitats, are permeable to reptiles and do not disrupt ecological connectivity at the landscape scale. This finding aligns with previous studies highlighting the potential of traditional agricultural systems or anthropogenic landscape features to support biodiversity by maintaining structural complexity and minimizing habitat fragmentation [13,42,52]. Nevertheless, the overall variability in the NCF values (min = 0.12; max = 2.33) of the olive groves underscores the heterogeneity of olive groves as habitats, with some patches offering more favorable conditions for reptile movement than others. This variability may be attributed to differences in management practices, understory vegetation, and proximity to natural habitats, factors which are supposed to influence both the current flow across the landscape and habitat suitability for reptiles [10,14,27,28].

### 4.2. Q2: When Embedded Within Natural Habitats, Do Olive Groves Provide the Same Connectivity as the Surrounding Natural Patches?

While significant differences in the normalized current flow (NCF) values between olive groves and some adjacent natural habitats (e.g., transitional woodlands/shrubs, sclerophyllous vegetation, and chestnut woodlands) were detected, it is important to note that the NCF values for the olive groves were consistently close to 1, indicating a diffuse current flow that was nearly identical to the connectivity provided by natural habitats. Moreover, the NCF values for olive groves and natural habitats were remarkably close to each other, suggesting that, in practical terms, olive groves offer connectivity levels that are functionally equivalent to those measured for the surrounding natural patches. This near-identical connectivity underscores the complementarity of olive groves in serving as connectivity pathways for reptiles. This finding aligns with previous studies that highlighted the role of traditional agricultural systems in supporting biodiversity by mimicking the structural and functional attributes of natural habitats; for example, the structural complexity of olive groves, including the presence of trees, shrubs, and ground cover, can facilitate reptile movement and habitat use [27,28]. Furthermore, the minor differences in the NCF values between olive groves and natural habitats may reflect subtle variations in habitat structure or microclimatic conditions, rather than a fundamental disparity in connectivity potential. For instance, natural habitats may offer a greater diversity of microhabitats, such as leaf litter, fallen logs, and rock crevices, which are critical for thermoregulation and sheltered ectotherms [53]. However, these differences are likely negligible in terms of their impact on overall landscape connectivity, particularly when olive groves are embedded within a matrix of natural habitats. This is consistent with the concept of “habitat complementarity”, according to which different habitat types collectively contribute to landscape connectivity by providing complementary resources, shelters, or movement pathways [54,55]. In conclusion, while olive groves are statistically distinct from some natural habitats in terms of NCF values, their practical contribution to ecological connectivity is nearly identical. This highlights the importance of olive groves as functional components of the landscape, particularly in regions where they are widespread and embedded within natural habitat patches, as is the case with the Liguria region.

### 4.3. Q3: Do Olive Groves Serve as a Part of the Ecological Corridors Connecting Protected Areas?

Our analysis of least-cost paths between protected areas revealed that olive groves play a significant role in facilitating connectivity for reptiles. In both western and eastern Liguria, a substantial proportion of least-cost paths (61.9% and 85.5%, respectively) intersected with olive groves. This highlights the importance of olive groves as stepping stones or corridors that link protected areas, enabling reptile movement across fragmented landscapes. These findings are particularly relevant in the context of conservation planning, as they underscore the potential of traditional agricultural systems to enhance connectivity between protected areas, thereby promoting gene flow and population viability [4,16], a key function already suggested in the Mediterranean basin for some artificial landscape features [3]. The high prevalence of olive groves in least-cost paths suggests that these agroecosystems are capable of bridging gaps between natural habitats and mitigating the effects of habitat fragmentation. This aligns with previous studies emphasizing the role of semi-natural habitats, such as hedgerows and man-made landscape elements in general, in fostering biodiversity in human-modified landscapes [11,14,21].

## 5. Conclusions

The results of this study have important implications for biodiversity conservation in agricultural landscapes. First, they highlight the value of traditional olive groves as habitats that support reptile movement and contribute to landscape connectivity. This underscores the need to preserve and promote low-intensity agricultural practices that maintain landscape structural complexity and minimize habitat fragmentation. Second, our findings suggest that olive groves can play a critical role in connecting protected areas, thereby enhancing the functionality of ecological networks. This is particularly relevant in the Mediterranean region, where the landscape is characterized by a dynamic mosaic of natural and human-modified elements [21,22]. Conservation strategies should therefore prioritize the integration of olive groves into regional connectivity plans, recognizing their potential to serve as ecological corridors.

However, it is important to note that not all olive groves are equally beneficial for biodiversity. The variability in the NCF values observed in this study suggests that the conservation value of olive groves may depend on their management practices and landscape context. For example, olive groves with dense understory vegetation and minimal chemical inputs are known to provide better habitat conditions for reptiles than intensively managed groves with bare soil and high pesticide use [13,30]. Future research should therefore evaluate the effectiveness of olive groves as connectivity pathways under varying management regimes, such as intensive versus traditional practices, and investigate the impact of land abandonment on their ecological function [19,56,57].

Finally, as the Mediterranean region, and the study area in particular, are currently facing ongoing or potential future changes in land use—such as land abandonment, land conversion for the development of infrastructures, or modifications in the cultivated crops—it is crucial to consider how these changes might affect habitat connectivity, particularly for vulnerable species or for those species at their distribution range limit. The fragmentation of habitats due to these transformations could alter or even impede the movement of these reptiles. We therefore recognize the importance of understanding these dynamics in future research and we suggest that subsequent studies focus on analyzing the impact of habitat transformations and land use changes on connectivity, especially for species on the periphery of their distribution range. Such studies would provide critical insights for conservation planning in dynamic landscapes and help mitigate the potential effects of habitat alteration on ecological networks.

## Figures and Tables

**Figure 1 animals-15-00909-f001:**
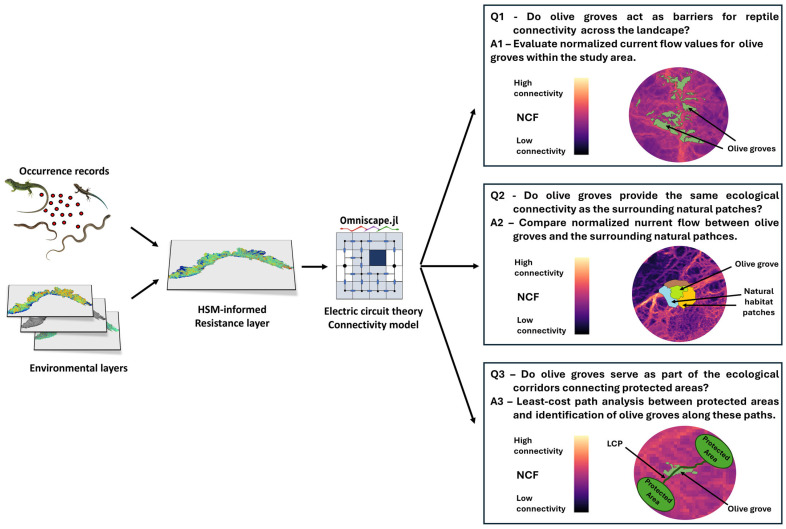
Graphical representation of the study design and framework. Q1–Q3 represent the three questions framed within the study, while A1–A3 represent the analyses performed to answer these questions.

**Figure 2 animals-15-00909-f002:**
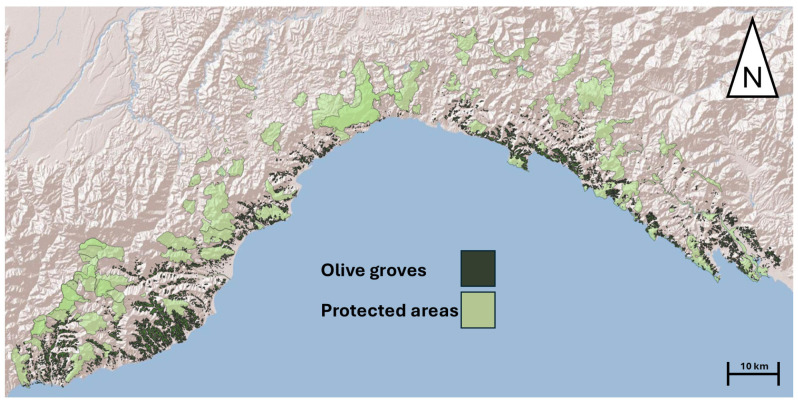
Map of the study area. Dark green polygons represent olive groves. Light green polygons represent protected areas.

**Figure 3 animals-15-00909-f003:**
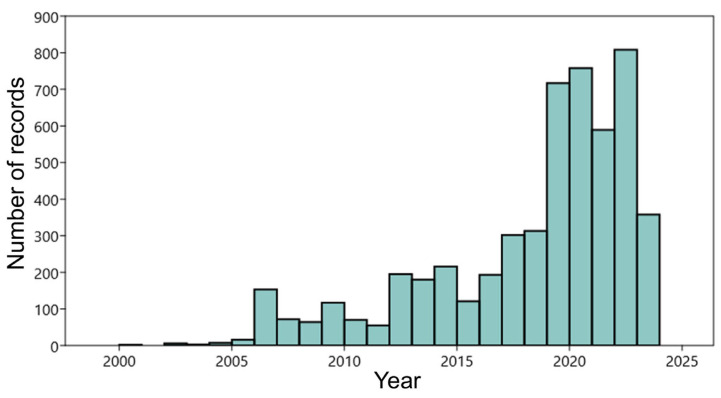
Temporal distribution of records included in the dataset, spanning from 2000 to 2024.

**Figure 4 animals-15-00909-f004:**
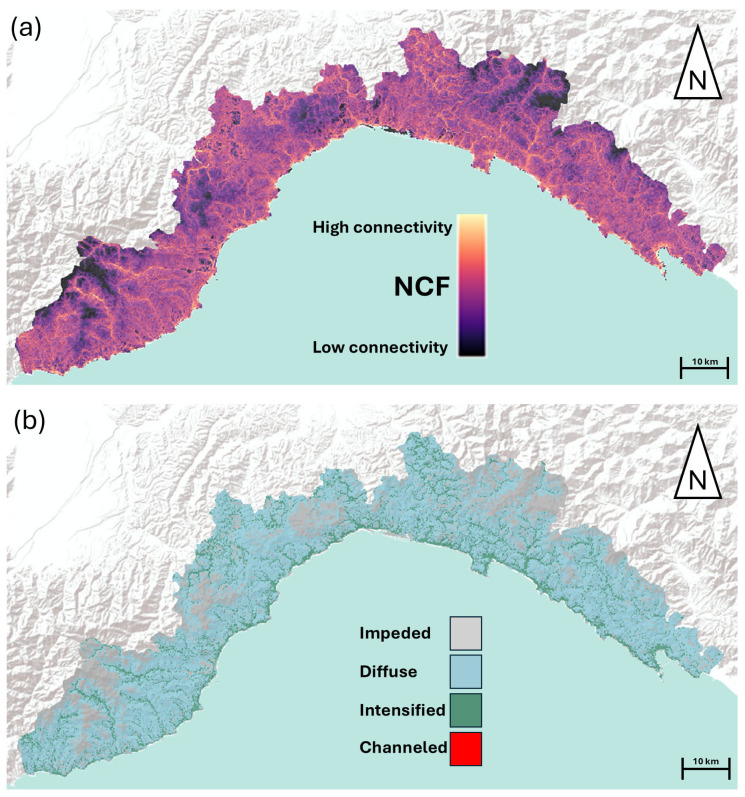
Maps of normalized current flow (NCF). (**a**) Continuous representation of NCF. (**b**) Categorized representation of NCF.

**Figure 5 animals-15-00909-f005:**
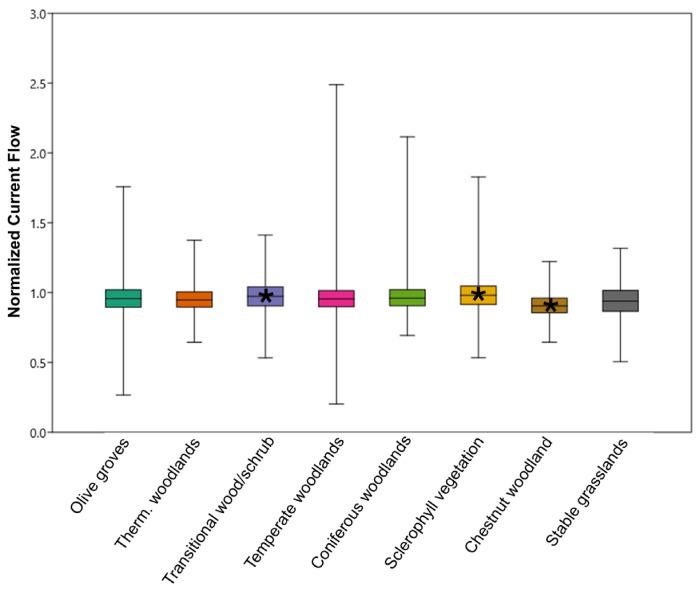
Boxplots representing NCF values for olive groves and neighboring natural vegetation patches. * Indicates a significant difference with respect to NCF for olive groves (Mann–Whitney test). Therm = Thermophilous.

**Table 1 animals-15-00909-t001:** List of reptile species included in the dataset, their IUCN category, the corresponding taxonomic level considered for the analysis, and the corresponding number of records included in the analysis.

Family	Species	IUCN Category	Taxon Considered in the Analyses	Number of Records
Gekkonidae	*Euleptes europaea* Genè, 1839	NT	*Euleptes europaea*	53
	*Hemidactylus turcicus* (Linnaeus, 1758)	LC	*Hemidactylus turcicus*	46
	*Tarentola mauritanica* (Linnaeus, 1758)	LC	*Tarentola mauritanica*	218
Scincidae	*Chalcides chalcides* (Linnaeus, 1758)	LC	*Chalcides* sp.	98
	*Chalcides striatus* (Cuvier, 1829)	LC		
Lacertidae	*Lacerta bilineata* Daudin, 1802	LC	*Lacerta bilineata*	1048
	*Podarcis muralis* Laurenti, 1768	LC	*Podarcis* sp.	2422
	*Podarcis siculus* Rafinesque, 1810	LC		
	*Timon lepidus* Daudin, 1802	LC	*Timon lepidus*	158
Anguidae	*Anguis veronensis* Pollini, 1818	LC	*Anguis veronensis*	126
Colubridae	*Coronella austriaca* Laurenti, 1768	LC	*Coronella* sp.	107
	*Coronella girondica* Daudin, 1803	LC		
	*Hierophis viridiflavus* (Lacépède, 1789)	LC	*Hierophis viridiflavus*	294
	*Malpolon monspessulanus* (Hermann, 1804)	LC	*Malpolon monspessulanus*	183
	*Natrix helvetica* (Linnaeus, 1758)	LC	*Natrix* sp.	281
	*Natrix maura* (Linnaeus, 1758)	LC		
	*Natrix tessellata* (Laurenti, 1768)	LC		
	*Zamenis longissimus* (Laurenti, 1768)	LC	*Zamenis longissimus*	102
Viperidae	*Vipera aspis* (Linnaeus, 1758)	VU	*Vipera aspis*	75

**Table 2 animals-15-00909-t002:** List of variables included in the HSMs, their description, and the origin of the layer.

Variable	Description	Origin of the Layer
Elevation a.s.l.	Meters above sea level	Derived from Digital Elevation Model (DEM)
TPI (Topographic Position Index)	An index giving information on the position of each cell within a slope. Assumes zero values when a cell lays on a topographic flat, negative values for valleys, and positive values for ridges.	Calculated from DEM using software SAGA (System for Automated Geoscientific Analyses, V 9.7.2)
Diffuse insolation	Index of diffuse insolation, expressing the amount of incoming solar radiation for each cell	Calculated from DEM using software SAGA for seasonal period of activity of reptiles (April–Sept)
Grasslands	Presence or absence of grasslands for each 100 m cell for the 2018 reference year	Provided by European Global Monitoring for Environment and Security Programme (COPERNICUS)
TCD (Tree Cover Density)	Percent density representing tree cover for each 100 m cell for the 2018 reference year	Provided by European Global Monitoring for Environment and Security Programme (COPERNICUS)
SWF (Small Woody Features)	Presence or absence of small woody vegetation for each 100 m cell for the 2018 reference year	Provided by European Global Monitoring for Environment and Security Programme (COPERNICUS)

**Table 3 animals-15-00909-t003:** List of natural habitat patches neighboring olive groves.

Patch Type	N of Polygons	% Coverage Area	Average NCF (SD)
Olive Groves	4212	-	0.96 (0.12)
Thermophilic mixed woodlands	1419	11.54	0.96 (0.07)
Transitional woodlands/shrubs	1059	11.37	0.97 (0.06)
Temperate mixed woodlands	852	9.18	0.96 (0.08)
Coniferous woodlands	717	5.84	0.97 (0.07)
Sclerophyllous vegetation	419	3.04	0.99 (0.08)
Chestnut woodlands	324	2.64	0.91 (0.08)
Stable grasslands	267	2.17	0.94 (0.06)

N of polygons: number of polygons neighboring olive groves for each habitat category; % coverage area: percent coverage area for each habitat category of the total area of neighboring habitat categories; Average NCF (SD): average and standard deviation of the NCF values for each natural habitat category neighboring olive groves.

## Data Availability

The raw data supporting the conclusions of this article will be made available by the authors on request.

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
