# Peer review of "Ecological Connectivity for Reptiles in Agroecosystems: A Case Study with Olive Groves in Liguria (Northwestern Italy)"

_animals, 2025, doi:10.3390/ani15070909_

Round 1
Reviewer 1 Report
Comments and Suggestions for Authors
I consider that the proposed article is a very good study that shows how the traditional olive grove can act as a connector and play an important role in maintaining the biodiversity of the reptile community
The study presented is very well structured, elaborated and clear in its description. The methodology used is adequate and the results very clear. The discussion is very well structured in parts, discussing separately the three questions raised at the beginning of the study. There are two small annotations in the manuscript that are more like suggestions.
Only one doubt has arisen during the reading of the study. I have not been able to see how the different mobility of the species has been considered when assessing the resistance of the different habitats. I would like to be answered although I consider that the article can be accepted with minimal changes.

Author Response
Reviewer #1 general comment: I consider that the proposed article is a very good study that shows how the traditional olive grove can act as a connector and play an important role in maintaining the biodiversity of the reptile community. The study presented is very well structured, elaborated and clear in its description. The methodology used is adequate and the results very clear. The discussion is very well structured in parts, discussing separately the three questions raised at the beginning of the study. There are two small annotations in the manuscript that are more like suggestions. Only one doubt has arisen during the reading of the study. I have not been able to see how the different mobility of the species has been considered when assessing the resistance of the different habitats. I would like to be answered although I consider that the article can be accepted with minimal changes.
Authors’ reply to general comment: We sincerely thank Reviewer #1 for his/hers valuable comments and positive feedback on our study. We have carefully addressed all the Reviewer’s suggestions and have revised the manuscript accordingly, as detailed in the following point-by-point response. Regarding the Reviewer’s question about how “different mobility has been considered,” we would like to clarify this as follows:
1)Species-specific habitat preferences and suitability: To account for variability in species’ responses to environmental characteristics or habitat features, we performed species-specific habitat suitability models to inform resistance layers. Using the same set of environmental variables but allowing for species-specific responses to these features, we captured differences in how each species interacts with its habitat. Based on these models, we built connectivity models for each species and then averaged landscape-scale connectivity to assess the importance of olive groves for the entire reptile community (L200-246). This approach aligns with systematic conservation planning principles, as it provides broader, community-level insights rather than species-specific guidelines, which are often challenging to implement at the landscape management scale.
2)Individual dispersal abilities: While species-specific movement preferences are incorporated through resistance surfaces (L191-206), the individual dispersal abilities of organisms within a species cannot be directly accounted for in connectivity modeling based on electric circuit theory. This kind of analysis/model “simply” evaluates how electric current flows between nodes (i.e. occurrence records of a given species) on a resistance surface (i.e. landscape friction map) and simulates animal movement using random walks—a stochastic process that creates a path by combining a series of random steps—along all possible movement routes. For this reason, specific dispersal abilities of individual organisms within a species are not accounted for. Instead, it generalizes movement patterns based on random walk dynamics, without considering variations in individual behavior or capacity. Instead, species-level movement ability is reflected through the use of species-specific resistance surfaces, which capture differences in how species perceive and navigate the landscape.
Reviewer #1 specific comments (PDF):
-L75: Also references 18 and 19
Authors: We added references 18 and 19 at this line
-L 78-82: I belive that this paragraph could bi insert after [4. 16]
Authors: we agree with the Reviewer and moved the paragraph accordingly.
-L92: added to indirect loss of prey
Authors: we agree with the Reviewer and added this part to the sentence
-L104: belive that the authors could add that the olive trees, specially old olive trees. offer a great number of refuges for reptiles
Authors: we acknowledge the Reviewer’s comment and modified the sentence to also include this concept.
-L171: The methodology used is correct but I have one question. How authors evaluate the differences in path resistance for each species, because the home range or dispersal capacity of each species are very different. I don't found the explanation of this variable.
Authors: Detailed reply to this comment has been given in the reply to the general comment.
Reviewer 2 Report
Comments and Suggestions for Authors
The impact of agriculture is the main cause of ongoing global biodiversity loss. At the same time, the main research is aimed at assessing the impact of intensive agriculture, where technologies aimed at increasing the productivity and yield of agricultural crops (use of pesticides and mineral fertilizers, monocultures) lead to landscape homogenization, creation of new habitats and changes in taxonomic composition (Lanz et al., 2018; Emmerson et al., 2016; Raven, Wagner, 2021). On the other hand, the impact of less intensive types of impacts on biodiversity has not been sufficiently studied. The authors presented an interesting and relevant study. However, practical use for the protection and maintenance of reptile populations, including those with a high threat status of population decline, is not presented. This will increase the relevance of the study and allow the authors' data to be used to develop measures.
General notes:
1. It is necessary to compare the taxonomic composition of reptiles taking into account the protection status and the "Number of record" depending on natural and to varying degrees transformed habitats, including historically formed olive groves.
Specific notes:
3. Lines 169-170 "Table 1. List of reptile taxa included in the dataset and the corresponding number of records included in the analysis." Formatting "sp." - is not highlighted in italics, taxa names should be complete.
2. Lines 163-166 "At the same time we considered genetically close species (e.g. Chalcides striatus and Chalcides chalcides) as a single entity at the genus taxonomic level. The resulting dataset consists of 5211 reptile records for 14 taxa, spanning from year 2000 to 2024 (Figure 3; Table 1).".
The authors indicate 14 taxa, and the region is Liguria, the study is distinguished by a unique composition of the reptile fauna, which exceeds that indicated by the authors and reaches at least 18 species, excluding turtles. I believe it is necessary to provide a complete list of taxa in Table 1, indicating the protection status in an additional column, as was indicated in the text for Chalcides sp. complex of at least 30 species, where it is indicated "(e.g. Chalcides striatus and Chalcides chalcides)". In Liguria, there is Chalcides striatus (Cuvier, 1829) - everywhere and the threatened species Chalcides chalcides (Linnaeus, 1758), list (LC IUCN). In this case, the following are also indicated:
Podarcis, sp.: Podarcis siculus (Rafinesque, 1810) and Podarcis muralis (Laurenti, 1768)
Coronella sp.: Coronella girondica (Daudin, 1803) and Coronella austriaca Laurenti, 1768
Natrix sp.: Natrix helvetica (Lacépède, 1789), Natrix maura (Linnaeus, 1758), Natrix tessellata (Laurenti, 1768).
Another factor, in addition to the protection status, are the limiting factors determining the distribution boundary and the features of the biotopic placement on the periphery of the range. The southern periphery for Coronella austriaca Laurenti, 1768, Hierophis viridiflavus (Lacépède, 1789). The northern periphery of the range of Coronella girondica (Daudin, 1803). The eastern periphery of the range - Timon lepidus (Daudin, 1802) and the western periphery of the range of Anguis veronensis Pollini, 1818.
There are also potentially invasive species (outside the study area) Euleptes europaea, Hemidactylus turcicus, Tarentola mauritanica, Podarcis siculus and Podarcis muralis, Natrix maura and Zamenis longissimus (Laurenti, 1768), as well as Vipera aspis (Linnaeus, 1758) having the IUCN status Vulnerable (IUCN 3.1.
A complete taxonomic composition, indicating the protection status, will confirm the importance of the study.
Author Response
Reviewer #2
Reviewer #2 general comment: The impact of agriculture is the main cause of ongoing global biodiversity loss. At the same time, the main research is aimed at assessing the impact of intensive agriculture, where technologies aimed at increasing the productivity and yield of agricultural crops (use of pesticides and mineral fertilizers, monocultures) lead to landscape homogenization, creation of new habitats and changes in taxonomic composition (Lanz et al., 2018; Emmerson et al., 2016; Raven, Wagner, 2021). On the other hand, the impact of less intensive types of impacts on biodiversity has not been sufficiently studied. The authors presented an interesting and relevant study. However, practical use for the protection and maintenance of reptile populations, including those with a high threat status of population decline, is not presented. This will increase the relevance of the study and allow the authors' data to be used to develop measures.
Authors’ reply to general comment: We express our gratitude to Reviewer#2 for his/hers comments and useful suggestions on our manuscript. We did out best to accommodate all the requests and replied to all specific questions raised by the Reviewer. Specifically, following Reviewer’s suggestion, we updated information on the taxonomic composition of the studied reptile community, also highlighting the conservation and protection status of the species included in the study. Detailed reply to Reviewer’s comments are listed below.
General notes:
1. It is necessary to compare the taxonomic composition of reptiles taking into account the protection status and the "Number of record" depending on natural and to varying degrees transformed habitats, including historically formed olive groves.
Authors: We thank the Reviewer for this valuable suggestion. In response, we have updated Table 1 to provide a more detailed taxonomic composition of reptiles at the species level, including the conservation/threat status for each species. This addition aligns with the Reviewer's recommendation to highlight the protection status of the taxa included in our study. Regarding the "Number of records" across different habitat types, we have carefully considered this suggestion. However, we decided not to include this information in the manuscript for the following reasons: The occurrence locations used in this study were not derived from extensive, systematic sampling across all habitat types. Therefore, reporting the number of records per habitat type could be misleading or uninformative, as it does not represent comprehensive coverage of the study area. The primary focus of this study is not to link the number of actual occurrences (probably flawed by detection and sampling bias) to a specific habitat, but rather to use the information embedded in these occurrence records to model connectivity in relation to the presence of historically formed olive groves. Including habitat-specific record counts could divert attention from the study's main objectives without providing meaningful insights into habitat preferences or usage.We hope this clarification addresses the Reviewer's concern while ensuring the manuscript remains focused on its core aims.
Specific notes:
Lines 169-170 "Table 1. List of reptile taxa included in the dataset and the corresponding number of records included in the analysis." Formatting "sp." - is not highlighted in italics, taxa names should be complete.
Authors: we apologize for this inconvenience, and we have corrected accordingly in the current version of the manuscript.
Lines 163-166 "At the same time we considered genetically close species (e.g. Chalcides striatus and Chalcides chalcides) as a single entity at the genus taxonomic level. The resulting dataset consists of 5211 reptile records for 14 taxa, spanning from year 2000 to 2024 (Figure 3; Table 1).".
The authors indicate 14 taxa, and the region is Liguria, the study is distinguished by a unique composition of the reptile fauna, which exceeds that indicated by the authors and reaches at least 18 species, excluding turtles. I believe it is necessary to provide a complete list of taxa in Table 1, indicating the protection status in an additional column, as was indicated in the text for Chalcides sp. complex of at least 30 species, where it is indicated "(e.g. Chalcides striatus and Chalcides chalcides)". In Liguria, there is Chalcides striatus (Cuvier, 1829) - everywhere and the threatened species Chalcides chalcides (Linnaeus, 1758), list (LC IUCN). In this case, the following are also indicated:
Podarcis, sp.: Podarcis siculus (Rafinesque, 1810) and Podarcis muralis (Laurenti, 1768)
Coronella sp.: Coronella girondica (Daudin, 1803) and Coronella austriaca Laurenti, 1768
Natrix sp.: Natrix helvetica (Lacépède, 1789), Natrix maura (Linnaeus, 1758), Natrix tessellata (Laurenti, 1768).
Another factor, in addition to the protection status, are the limiting factors determining the distribution boundary and the features of the biotopic placement on the periphery of the range. The southern periphery for Coronella austriaca Laurenti, 1768, Hierophis viridiflavus (Lacépède, 1789). The northern periphery of the range of Coronella girondica (Daudin, 1803). The eastern periphery of the range - Timon lepidus (Daudin, 1802) and the western periphery of the range of Anguis veronensis Pollini, 1818.
There are also potentially invasive species (outside the study area) Euleptes europaea, Hemidactylus turcicus, Tarentola mauritanica, Podarcis siculus and Podarcis muralis, Natrix maura and Zamenis longissimus (Laurenti, 1768), as well as Vipera aspis (Linnaeus, 1758) having the IUCN status Vulnerable (IUCN 3.1.
A complete taxonomic composition, indicating the protection status, will confirm the importance of the study.
Authors: The Reviewer is right, our taxonomic list reported in Table 1 was incomplete and misleading in the first version of the manuscript. Now we completely reframed Table 1 following Reviewer’s comments and reported full species list, the traxonmic level at which the analysis has been conducted, the number of occurrence records employed and the conservation status of each species. Additionally, regarding species with range peripheries within the study area, we have added a clarifying sentence in the manuscript to explain how we addressed this issue. This addition aims to provide further context and ensure that our methodology is clear and comprehensive.
Reviewer 3 Report
Comments and Suggestions for Authors
The manuscript by Costa et al. presents the results of an interesting, well-done study with meaningful implications, not just for the conservation of Mediterranean reptiles, but for conservation and landscape management in general. I have some suggestions to improve the English, but other than that, I believe this paper will be a significant contribution.
Comments on the Quality of English LanguageGenerally, the authors need to use tenses consistently. For example, in lines 105 and 296, they used present tense in describing the study, but elsewhere (e.g., 325), they used past tense. A good rule of thumb is to use past tense for completed events, including what they found in their study, and use present tense only for broad, ongoing patterns.
The references need to be consistent. For example, #5 (line 435) should not be in all caps. Journal abbreviations have periods in line 524 but not in line 444. Also, journal titles for many references are not abbreviated when they should be.
Specific corrections follow line numbers below:
43 change “it’s” to “its” as possessive
45 change “agricultural expansion” to “agriculture and its associated expansion”
49 end this sentence with “…communities [3-5].” Start a new sentence with “Ecological connectivity impacts the survival…”
50 delete comma; insert “those” before “relying”
65 insert “an” before “artificial”
69 capitalize “Mediterranean”
78 delete comma
100 change “to sustain” to “of sustaining”
101 change “productions” to “practices”; change “are recognized” to “being recognized”
102 change “reptiles and may play a crucial role” to “reptiles, and possibly playing a crucial role”
103 replace “their contribution” with “the contribution of olive groves”
126 add comma after “scale”
130 delete comma
147 delete comma
204 replace “performing” with “performative”
229 replace “because it biased” with “which can be biased”
263 remove hyphen from “ecological”
279 delete “For what concerns variable importance” and begin sentence with “Elevation above…”
280 replace “resulted the” with “were”
286 insert “a” before “few”
299 change “are” to “is” [subject is “movement,” not “categories”]
302 change “significantly different” to “in significant differences”
323 insert “the” before “first”
333,338,343,359 remove italics from reference numbers
366 no need to use both quotation marks and italics to underscore the importance of habitat complementarity; use one or the other
Author Response
Reviewer #3
Reviewer #3 general comment: The manuscript by Costa et al. presents the results of an interesting, well-done study with meaningful implications, not just for the conservation of Mediterranean reptiles, but for conservation and landscape management in general. I have some suggestions to improve the English, but other than that, I believe this paper will be a significant contribution.
Authors’ reply to general comment: We are grateful to Reviewe#3 for his/her comments and suggestions on our study. We followed all his comments and amended the manuscript accordingly.
Comments on the Quality of English Language:Generally, the authors need to use tenses consistently. For example, in lines 105 and 296, they used present tense in describing the study, but elsewhere (e.g., 325), they used past tense. A good rule of thumb is to use past tense for completed events, including what they found in their study, and use present tense only for broad, ongoing patterns.
Authors: We appreciate the Reviewer’s suggestion on the English language of our manuscript. We have carefully addressed the issue of inconsistent tense usage by revising the text to ensure that past tense is used for completed events (e.g., methods and findings) and present tense is reserved for general statements and ongoing patterns. Additionally, to further enhance the quality of the language, we enlisted the assistance of a native English speaker to thoroughly review and edit the manuscript. We hope that these revisions have improved the readability and overall clarity of the manuscript.
Reviewer #3: The references need to be consistent. For example, #5 (line 435) should not be in all caps. Journal abbreviations have periods in line 524 but not in line 444. Also, journal titles for many references are not abbreviated when they should be.
Authors: we double-checked and corrected the reference list.
Specific corrections follow line numbers below:
L43 change “it’s” to “its” as possessive
Authors: changed as suggested
L45 change “agricultural expansion” to “agriculture and its associated expansion”
Authors: changed
L49 end this sentence with “…communities [3-5].” Start a new sentence with “Ecological connectivity impacts the survival…”
Authors: changed as suggested
L50 delete comma; insert “those” before “relying”
Authors: deleted as suggested
L65 insert “an” before “artificial”
Authors: changed as suggested
L69 capitalize “Mediterranean”
Authors: done
L78 delete comma
Authors: deleted as suggested
L100 change “to sustain” to “of sustaining”
Authors: changed as suggested
L101 change “productions” to “practices”; change “are recognized” to “being recognized”
Authors: changed as suggested
L102 change “reptiles and may play a crucial role” to “reptiles, and possibly playing a crucial role”
Authors: changed as suggested
L103 replace “their contribution” with “the contribution of olive groves”
Authors: changed as suggested
L126 add comma after “scale”
Authors: added
L130 delete comma
Authors: deleted
L147 delete comma
Authors: deleted
L204 replace “performing” with “performative”
Authors: replaced as suggested
L229 replace “because it biased” with “which can be biased”
Authors: changed as suggested
L263 remove hyphen from “ecological”
Authors: removed as suggested
L279 delete “For what concerns variable importance” and begin sentence with “Elevation above…”
Authors: deleted as suggested
L280 replace “resulted the” with “were”
Authors: replaced as suggested
L286 insert “a” before “few”
Authors: changed as suggested
L299 change “are” to “is” [subject is “movement,” not “categories”]
Authors: changed as suggested
L302 change “significantly different” to “in significant differences”
Authors: changed as suggested
L323 insert “the” before “first”
Authors: done
L333,338,343,359 remove italics from reference numbers
Authors: done
L366 no need to use both quotation marks and italics to underscore the importance of habitat complementarity; use one or the other
Authors: done
Round 2
Reviewer 2 Report
Comments and Suggestions for Authors
The authors have made corrections. The manuscript has been improved. But there are factual comments.
Comments:
1. Lines: 198-199, Table 1. List of reptile species included in the dataset, their IUCN category, the corresponding taxonomic level considered for the analysis and the corresponding number of records included in the analysis.
Please clarify the protection status for
- Euleptes europaea, listed as "LC", listed as "NT", https://www.iucnredlist.org/species/61446/12486542
- Chalcides striatus, listed as "NT", but listed as "LC", https://www.iucnredlist.org/species/61489/86452339
- Vipera aspis, listed as "VU", https://www.iucnredlist.org/species/61591/137859549
- Timon lepidus, listed as "LC", https://www.iucnredlist.org/species/218293375/137858480
2. Ibid., "sp." - text format, without italics.
3. Ibid., full species names, with the author of the description, for example, "Chalcides striatus (Cuvier, 1829)".
4. Line: 408, "[53]", link without italics.
5. Analysis of the connection with habitats is necessary if their transformation is underway (in the process) or is planned, with the development of transport infrastructure or a change in cultivated crops. Thus, for vulnerable species (with IUCN status) and on the periphery (border) of habitats in Liguria, additional information related to limiting factors in the study area is necessary.
Author Response
Reviewer #2: The authors have made corrections. The manuscript has been improved. But there are factual comments.
Authors: We are grateful to the Reviewer for his/hers valuable comments on the revised manuscript. In this new version we have amended the manuscript accordingly to the useful suggestion provided. Detailed reply to Reviewer’s comments are reported below.
Comments:
1. Lines: 198-199, Table 1. List of reptile species included in the dataset, their IUCN category, the corresponding taxonomic level considered for the analysis and the corresponding number of records included in the analysis.
Please clarify the protection status for
- Euleptes europaea, listed as "LC", listed as "NT", https://www.iucnredlist.org/species/61446/12486542
- Chalcides striatus, listed as "NT", but listed as "LC", https://www.iucnredlist.org/species/61489/86452339
- Vipera aspis, listed as "VU", https://www.iucnredlist.org/species/61591/137859549
- Timon lepidus, listed as "LC", https://www.iucnredlist.org/species/218293375/137858480
Authors: we apologize for the inconvenience, it was a typo error occurred while completing the table. Now we have checked and properly corrected the table.
Ibid., "sp." - text format, without italics.
Authors: Changed as suggested
3. Ibid., full species names, with the author of the description, for example, "Chalcides striatus (Cuvier, 1829)".
Authors: Changed here and elsewhere in the table, as suggested
4. Line: 408, "[53]", link without italics.
Authors:changed as suggested
5. Analysis of the connection with habitats is necessary if their transformation is underway (in the process) or is planned, with the development of transport infrastructure or a change in cultivated crops. Thus, for vulnerable species (with IUCN status) and on the periphery (border) of habitats in Liguria, additional information related to limiting factors in the study area is necessary.
Authors: We appreciate the Reviewer’s comment regarding the potential impact of habitat transformation and land use changes on the connectivity of habitat patches, particularly for the reptile community and, more broadly, for vulnerable species. While we acknowledge that such transformations could indeed alter habitat connectivity, addressing these patterns is beyond the scope of the current study. Additionally, implementing a thorough analysis of this aspect would be challenging with the data currently available and the existing environmental layers for the study area. Nevertheless, recognizing the importance of this issue, we have added a sentence in the discussion section to highlight this concern and suggest directions for future research in this field. Specifically we added the following sentence in the conclusions section: “ Finally, as the Mediterranean region, and the study area in particular, are currently facing ongoing or potential future changes in land use —such as land abandonment, land conversion for the development of infrastructures or the transformation of cultivated crops— it is crucial to consider how these changes might affect habitat connectivity, par-ticularly for vulnerable species or for those species at the distribution range limit. The fragmentation of habitats due to these transformations could alter or even impede the movement of these reptiles. We therefore recognize the importance of understanding these dynamics in future research, and we suggest that subsequent studies focus on analyzing the impact of habitat transformations and land use changes on connectivity, especially for species on the periphery of their distribution range. Such studies would provide critical in-sights for conservation planning in dynamic landscapes and help mitigate the potential effects of habitat alteration on ecological networks.”
Round 3
Reviewer 2 Report
Comments and Suggestions for Authors
Thank you for making the changes. The actual errors have been fixed.